# Enzymatic Transglycosylation Features in Synthesis of 8-Aza-7-Deazapurine Fleximer Nucleosides by Recombinant *E. coli* PNP: Synthesis and Structure Determination of Minor Products

**DOI:** 10.3390/biom14070798

**Published:** 2024-07-04

**Authors:** Barbara Z. Eletskaya, Anton F. Mironov, Ilya V. Fateev, Maria Ya. Berzina, Konstantin V. Antonov, Olga S. Smirnova, Alexandra B. Zatsepina, Alexandra O. Arnautova, Yulia A. Abramchik, Alexander S. Paramonov, Alexey L. Kayushin, Anastasia L. Khandazhinskaya, Elena S. Matyugina, Sergey N. Kochetkov, Anatoly I. Miroshnikov, Igor A. Mikhailopulo, Roman S. Esipov, Irina D. Konstantinova

**Affiliations:** 1Shemyakin-Ovchinnikov Institute of Bioorganic Chemistry, Russian Academy of Sciences, Moscow 117997, Russia; fraubarusya@gmail.com (B.Z.E.); anton-mironov1999@inbox.ru (A.F.M.); ifateev@gmail.com (I.V.F.); berzina_maria@mail.ru (M.Y.B.); gescheites@gmail.com (O.S.S.); alex.zatsepina@gmail.com (A.B.Z.); 8818818@mail.ru (A.O.A.); ugama@yandex.ru (Y.A.A.); a.s.paramonov@gmail.com (A.S.P.); kayushin.alexej@yandex.ru (A.L.K.); esipov@ibch.ru (R.S.E.); 2Institute of Biochemical Technology and Nanotechnology, Peoples’ Friendship University of Russia Named after Patrice Lumumba, Miklukho-Maklaya St. 6, Moscow 117198, Russia; 3Engelhardt Institute of Molecular Biology, Russian Academy of Sciences, 32 Vavilov St., Moscow 119991, Russia; khandazhinskaya@bk.ru (A.L.K.); matyugina@gmail.com (E.S.M.); snk1952@gmail.com (S.N.K.); 4Institute of Bioorganic Chemistry, National Academy of Sciences, Acad. Kuprevicha 5/2, 220141 Minsk, Belarus; imikhailopulo@gmail.com

**Keywords:** fleximer base, recombinant *E. coli* PNP, catalytic site, molecular modeling

## Abstract

Enzymatic transglycosylation of the fleximer base 4-(4-aminopyridine-3-yl)-1H-pyrazole using recombinant *E. coli* purine nucleoside phosphorylase (PNP) resulted in the formation of “non-typical” minor products of the reaction. In addition to “typical” N1-pyrazole nucleosides, a 4-imino-pyridinium riboside and a N1-pyridinium-N1-pyrazole bis-ribose derivative were formed. N1-Pyrazole 2′-deoxyribonucleosides and a N1-pyridinium-N1-pyrazole bis-2′-deoxyriboside were formed. But 4-imino-pyridinium deoxyriboside was not formed in the reaction mixture. The role of thermodynamic parameters of key intermediates in the formation of reaction products was elucidated. To determine the mechanism of binding and activation of heterocyclic substrates in the *E. coli* PNP active site, molecular modeling of the fleximer base and reaction products in the enzyme active site was carried out. As for N1-pyridinium riboside, there are two possible locations for it in the PNP active site. The presence of a relatively large space in the area of amino acid residues Phe159, Val178, and Asp204 allows the ribose residue to fit into that space, and the heterocyclic base can occupy a position that is suitable for subsequent glycosylation. Perhaps it is this “upside down” arrangement that promotes secondary glycosylation and the formation of minor bis-riboside products.

## 1. Introduction

Purine nucleoside phosphorylase (PNP) (EC 2.4.2.1, https://www.brenda-enzymes.org/enzyme.php?ecno=2.4.2.1 (accessed on 15 May 2024)) is of particular interest for biocatalytic nucleoside synthesis due to its broad substrate specificity [1,2,3,4,5,6,7,8,9,10]. In addition to natural purine nucleosides such as adenosine, guanosine, and inosine, *Escherichia coli* purine nucleoside phosphorylase [11,12,13] can also recognize substrates with substitutions at positions 2 and 6 on the purine cycle [14]. Bases and nucleosides with halogen atoms (chlorine and fluorine) at the C2 position of purine are also suitable substrates [15,16,17]. Neither small substituents in the C6 position (for example, halogens and a methoxy group [18]), nor larger ones, up to cycloalkyl fragments [19,20] and fluorinated benzoxazines attached to the C6 atom through a hexanoyl spacer [21] affect the ability of the enzyme to glycosylate them at the N9 position of purine (Figure 1).

Additionally, the enzyme is able to interact with aza(deaza)-modified bases [22]. In such heterocyclic structures, the nitrogen atom at position 7 of the purine cycle can be substituted with a carbon atom (CH-group), while the carbon atoms at positions 5 and 8, on the contrary, can be replaced with nitrogen atoms [23,24,25].

The enzyme has a unique ability to interact with non-typical bases, whose structure significantly differs from natural substrates. Derivatives of 1,2,4-triazoles [26,27], benzimidazoles [28,29,30], and imidazoles [31,32] proved to be suitable substrates for PNP. The recent discovery that imidazoles substituted at position 4 were substrates for PNP has led to the development of a novel enzymatic method for the synthesis of so-called “fleximeric” nucleosides [32,33]. Tricyclic bases also appear to be substrates for *E. coli* PNP [32,34,35,36]. All the data presented above indicate the remarkable adaptability of the active site of the enzyme, which can catalyze reactions with various bases and nucleosides that significantly differ from the natural substrates.

Regarding the carbohydrate part of the molecule, *E. coli* PNP also demonstrates tolerance to nucleosides containing D-arabinose [37], 3-deoxyribose [38], and 2-deoxy-2-fluoroarabinose [39] residues. It is especially important to note that the activity of *E. coli* PNP against purine arabinosides is significantly greater than the activity of mammalian PNP [40].

To date, enzymatic transglycosylation of modified heterocyclic bases has been actively used in the synthesis of various new nucleosides, due to the stereo- and regiospecificity of the method [15,41]. However, the principle of regioselectivity may not always be completely achieved: Figure 2 shows examples of heterocycles, the glycosylation of which forms isomers with different positions of the glycoside bond.

Earlier research into the possibility of using bacterial *E. coli* PNP for the transglycosylation of 3-deazapurine **1** has indicated the formation of N9 and N7 isomers [25]. The formation of N7, N9-regioisomers has also been observed during glycosylation of N2-acetylguanine **2** [14,42]. It has been shown that recombinant PNP from calf spleen catalyzes the ribosylation of 2,6-diamino-8-azapurine **3** mainly forming N7 and N8 nucleosides [43]. On the other hand, *E. coli* PNP gave a mixture of N8 and N9 isomers.

The formation of N1, N3-imidazole isomers has also been reported in several studies [29,44] of enzymatic transglycosylation of 5-substituted 4,6-difluorobenzoimidazoles **4**. A unique case of enzymatic transglycosylation of the aromatic amino group of 2-amino-1,3-benzoxazole **5** was reported by the same authors [45]. Interestingly, the resulting riboside and 2′-deoxyriboside of 2-amino-1,3-benzoxazole 5 spontaneously rearrange into the corresponding β-D-pyranosides **6**.

Usually, glycosylation of substituted 1,2,4-triazoles occurs at the N1 position of the triazole cycle. However, when ribosylation is performed on 5-(benzyloxymethyl)-1H-1,2,4-triazole-3-carboxamide **7**, in addition to the main N1-glycosylated product (80%), the formation of minor N2- and N4-isomers was also observed (10% each) [26]. It should be mentioned that the minor glycosylation products of base **7** were synthesized under conditions using high concentrations of PNP in reactions that were carried out for a long time (360 h).

Such variability of glycosylation sites is due to the similar nucleophilicity of nitrogen atoms in the heterocyclic base. Non-selective glycosylation was observed not only for PNP, but also for another type of enzyme from the glycosyltransferases family—N-deoxyribosyltransferases (NDTs, EC 2.4.2.6, https://www.brenda-enzymes.org/enzyme.php?ecno=2.4.2.6 (accessed on 15 May 2024)) from *Lactobacillus*. Both NDT *Lactobacillus* and *E. coli* PNP catalyze the formation of two reaction products—N1 and N7 oxoimidazopurine **8** nucleosides, and N1 and N3 oxoimidazopurine **9** nucleosides [34]. In the case of enzymatic transglycosylation of tricyclic base **10** using NDT *Lactobacillus leichmannii*, the formation of three products was observed: two N1′ and N3′ mono-glycosides and N1,N1′-bis-glycoside; in the case of *E. coli* PNP, glycosylation occurs only at positions N1′ or N3′ of the imidazole [32]. In the case of glycosylation of 4-substituted imidazoles **11** by *L. Leichmanni* NTD*,* the lack of regio-specificity toward N1 and N3 nitrogen in the imidazole cycle allows *distal* and proximal fleximer 2′-deoxyribosides to be obtained [32].

Overall, *E. coli* PNP demonstrates efficient biocatalytic properties with high regio- and stereoselectivity. At the same time, testing the activity of an ever-increasing number of natural heterocyclic bases analogs leads to the production of structurally isomeric nucleosides. We have recently described an enzymatic method for the synthesis of ribo- and 2′-deoxyribonucleosides of new fleximer bases **12–14** (Figure 3) [46]. A two-enzyme approach using *E. coli* UP and PNP was used to perform the transglycosylation reaction of fleximer bases. Uridine phosphorylase (*E. coli* UP) was used to generate ribose-1-phosphate and 2′-deoxyribose- 1-phosphate from Urd and dUrd, respectively.

In the case of glycosylation of 4-(4-aminopyridin-3-yl)-1H-pyrazole **12** using recombinant *E. coli* PNP, several reaction products were obtained. In order to understand the possible pathways of substrate **12** binding and activation in the catalytic center of *E. coli* PNP, we performed a quantum chemical analysis of the initial structures of the heterocycles and the final nucleotides. We also performed computer modeling of the interaction between base **12** and the active site of the enzyme in order to better understand how the formation of various nucleoside products becomes possible. The results of the study are discussed in detail in this paper.

## 2. Materials and Methods

### 2.1. General Information

All commercially available chemical materials (Acros Organics (Geel, Belgium), Sigma-Aldrich (Burlington, MA, USA), and Fluka (Buchs, Switzerland)) were obtained from commercial suppliers and used without any purification. Fleximer base **12** was obtained using the method previously published [46].

Plasmid vectors containing the genes encoding EcoPNP and EcoUP constructed in previous research were used for the transformation of competent *E. coli* ER2566 (New England Biolabs) strain cells [47]. The resulting producer strains were cultivated at 37 °C in the Luria–Bertani medium (10 g. tryptone, 5 g. yeast extract 10 g. NaCl per 1 L) with 100 µg/mL ampicillin. Upon reaching the optical density of A^595^ = 0.8, the cell cultures were supplemented with 0.4 mM of IPTG, followed by further cultivation of 4 h at 37 °C, which allowed to obtain enzymes in soluble form.

The cell biomass was harvested through centrifugation and resuspended (1:10 *w*/*v*) in buffer containing 50 mM Tris-HCl pH 8.0, 5 mM EDTA, 1 mM benzydamine hydrochloride, and 1 mM PMSF. The cell suspension was subjected to ultrasonic disintegration and centrifugation to remove the cell debris. The resulting supernatant was applied onto a column packed with Q Sepharose FF sorbent (Cytiva, Washington, DC, USA), equilibrated with buffer containing 50 mM Tris-HCL pH 8.0, 5 mM EDTA. The proteins were eluted with a linear gradient 0–500 mM NaCl. The fractions containing the desired enzymes were pooled, concentrated on the YM-30 membrane (Merck Millipore, Darmstadt, Germany), and applied onto columns packed with Superdex 200 (GE Healthcare, Chicago, IL, USA) sorbent, pre-equilibrated the final buffer: 50 mM Tris-HCl pH 8.0, 200 mM NaCl. 0.04% NaN_3_. Fractions containing the target enzymes were pooled and concentrated on the YM-30 membrane (Merck Millipore, Burlington, MA, USA) to a concentration of approximately 30 mg/mL. The resulting aliquots of the enzyme’s solution were stored at −80 °C for further experiments. We measured the protein concentration according to Bradford and determined the enzyme purity through SDS PAGE. Solutions of recombinant *E. coli* UP and PNP in 10 mM potassium phosphate buffer (pH 7.0) with activities 1700 and 1400 U per mL, respectively, were used.

High-performance liquid chromatography (HPLC) was performed on the Waters system (1525 Binary HPLC Pump, 2489 UV/VIS Detector, Breeze v2 software, Waters Inc., Milford, MA, USA) using: Ascentis^®^ Express C18, 2.7 μm (7.5 × 3.0 mm) column. Eluent A: water + 0.1% TFA. Eluent B: 70% acetonitrile: 30% water + 0.1% TFA, injection 10 μL, detection at 225 nm and 280 nm. Gradient program: 0%B from 0 to 5 min, then 0–10%B from 5 to 15 min.

NMR spectra were recorded on Bruker Avance II 700 spectrometers (Bruker BioSpin, Rheinstetten, Germany) in DMSO-d_6_ at 30 °C. The operating frequencies for ^1^H, ^13^C, and ^15^N were 700 MHz, 176 MHz, and 71 MHz, respectively. Chemical shifts in ppm (δ) were measured relative to the residual solvent signal as internal standard (2.50). Coupling constants (*J*) were measured in Hz.

High-resolution mass spectra (HRMS) were registered on a Bruker Daltonics microTOF-Q II instrument using electrospray ionization (Bruker Daltonics, Bremen, Germany). The measurements were acquired in a negative ion mode with the following parameters: interface capillary voltage—3700 V; mass range from *m*/*z* 50 to 3000; external calibration (Electrospray Calibrant Solution, Fluka, Buchs, Switzerland); nebulizer pressure—0.3 Bar; flow rate—3 µL/min; dry gas nitrogen (4.0 L/min); interface temperature was set at 180 or 190 °C. A syringe injection was used. The UV spectra were recorded on the Agilent 6224, ESI-TOF, LC/MS (Agilent, Santa Clara, CA, USA) in water.

### 2.2. Enzymatic Reactions

Enzymatic transglycosylation was performed under the following general conditions. The flex-base **12** (160.18 g/mol, 0.312 µmol) and uridine (244.2 g/mol)/2′-deoxyuridine (228.2 g/mol) at a ratio of 1:10 were dissolved in 156 mL 10 mM potassium phosphate buffer (pH 7.0) at 40–50 °C. The enzymes 234 U PNP and 125 U UP *E. coli* were added. The reaction mixtures were incubated at 50 °C until the conversion reached **16**—49% (120 h), **17**—46% (1368 h), and **19**—53% (21 h) according to the RP-HPLC data. Conversion was calculated using HPLC data as the ratio of the peak area of the product to the sum of the peak areas of the product and the starting base. The ribosylation reaction was performed in a single vessel, after which the reaction mixture was separated into two portions. Product 16 was isolated from the first portion, and the reaction continued in the second portion until maximum conversion of product **17** had been achieved.

### 2.3. Isolation of Enzymatic Glycosylation Products

When conversion reached the maximal value, the reaction was terminated by ultrafiltration (MWCO 10 K) Macrosep^®^ (PALL Life Sciences, East Hills, NY, USA). The products were purified by reversed-phase column chromatography (silica gel C18, Merck, Darmstadt, Germany), column 150 × 20 mm. Nucleosides were eluted from a column with gradient of acetonitrile in water (250 mL, flow rate 1 mL/min). Then, minor nucleosides were eluted by acetonitrile + 0.005% TFA (100 mL, flow rate 1 mL/min). Fractions were neutralized immediately with aqueous ammonia and lyophilized (Figure 4). 

1-(β-D-ribofuranosyl)-3-(pyrazol-4-yl)-4-iminopyridine (**16**) ^1^H NMR: δ = 13.34 (s, 1H, H-1A), 8.55 (br.sign, 0.72H, NH), 8.55 (d, *J* = 1.6, 1H, H-2B), 8.35 (dd, *J* = 1.5, 7.3, 1H, H-6B), 8.13 (br.sign, 0.84H, H-5A), 7.80 (br.sign, 0.85H, H-3A), 7.42 (br.sign, 0.62H, NH), 7.00 (d, *J* = 7.3, 1H, H-5B), 5.67 (d, *J* = 5.6 1H, H-1′), 5.62 (d, *J* = 6.1, 1H, 2′-OH), 5.35 (d, *J* = 4.3, 1H, 3′-OH), 5.31 (t, *J* = 4.8, 1H, 5′-OH), 4.16 (m, 1H, H-2a′), 4.09 (m, 1H, H-3′), 4.08 (m, 1H, H-4′), 3.73 (m, 1H, Ha-5′), 3.66 (m, 1H, Hb-5′). ^13^C NMR: δ = 157.17 (C4B), 137.83 (C3A), 137.59 (C2B), 137.59 (C6B), 127.93 (C5A), 114.63 (C3B), 111.05 (C4A), 109.43 (C5B), 97.28 (C1′), 86.99 (C4′), 76.45 (C2′), 70.15 (C3′), 60.62 (C5′).^15^N NMR: δ = 179.6 (N1B), 99.0 (NH_2_). Yield 3 mg (17%). Purity 96.65% (t_R_ 6.4 min). HRMS, *m*/*z*: calculated for C_13_H_17_N_4_O_4_ [M]^+^ 293.1244, found [M]^+^ 293.1234. UV λ max 289 nm

Bis-riboside (**17**) ^1^H NMR: δ = 8.58 (d, *J* = 1.7, 1H, H-2B), 8.53 (br.sign, 0.5H, NH), 8.39 (dd, *J* = 1.7, 7.4, 1H, H-6B), 8.29 (s, 1H, H-5A), 7.82 (s, 1H, H-3A), 7.41 (br.sign, 0.5H, NH), 6.98 (d, *J* = 7.3, 1H, H-5B), 6.15 (t, *J* = 6.3 1H, H-1′-C), 6.10 (t, *J* = 6.4, 1H, H-1′-D), 5.41 (d, *J* = 4.0, 1H, 3′-OH-D), 5.28 (d, *J* = 4.3, 1H, 3′-OH-C), 5.24 (t, *J* = 4.9, 1H, 5′-OH-D), 4.82 (t, *J* = 5.5, 1H, 5′-OH-C), 4.40 (m, 1H, H-3′-C), 4.35 (m, 1H, H-3′-D), 3.97 (q, *J* = 3.3; 7.5, 1H, H-4′-D), 3.86 (m, 1H, H-4′-C), 3.68 (m, 1H, HC-5′-D), 3.63 (m, 1H, Hb-5′-D), 3.55 (m, 1H, Ha-5′-C), 3.46 (m, 1H, Hb-5′-C), 2.65 (m, 1H, Ha-2′-C), 2.42 (m, 1H, Ha -2′-D), 2.33 (m, 1H, Hb-2′-D), 2.28 (m, 1H, Hb-2′-C). ^13^C NMR: δ = 156.95 (C4B), 138.56 (C3A), 138.39 (C2B), 137.63 (C6B), 128.97 (C5A), 113.82 (C3B), 112.15 (C4A), 109.50 (C5B), 94.18 (C1′-D), 89.25 (C1′-C), 88.89 (C4′-D), 87.83 (C4′-C), 70.64 (C3′-D), 69.88 (C3′-C), 61.95 (C5′-C), 60.70 (C5′-D), 41.92 (C2′-D), 39.94 (C2′-C). ^15^N NMR: δ = 303.4 (N2A), 228.0 (N1A), 184.6 (N1B). Yield 3 mg (11%). Purity 98.28% (t_R_ 6.7 min). HRMS, *m*/*z*: calculated for C_18_H_25_N_4_O_8_ [M]^+^ 425.1667, found [M]^+^ 425.1648. UV λ max 289 nm.

Bis-2′-deoxyriboside (**19**) ^1^H NMR: δ = 8.59 (br.sign, 0.46 H, NH), 8.55 (d, *J* = 1.6, 1H, H-2B), 8.37 (dd, *J* = 7.6, 1H, H-6B), 8.33 (s, 1H, H-5A), 7.84 (s, 1H, H-3A), 7.47(br.sign, 0.59H, NH), 7.01 (d, *J* = 7.4, 1H, H-5B), 5.73 (d, *J* = 4.0, 1H, H-1′-C), 5.67 (d, *J* = 5.5, 1H, H-1′-D), 5.64 (br.sign, 0.77 H, 2′-OH-D), 5.41(br.sign,1H, 2′-OH-C), 5.37 (br.sign, 1 H, 3′-OH-D), 5.33 (br.sign, 2 H, 5′-OH-D), 5.16 (br.sign, 1 H, 3′-OH-C), 4.85 (br.sign, 1 H, 1 H, 5′-OH-C), 4.40 (m, 1 H, H-2′-C), 4.17 (m, 1 H, H-3′-C), 4.16 (m, 1 H, H-2′-D), 4.07 (m, 1H, H-3′-D), 4.08 (m, 1 H, H-4′-D), 3.94 (m, 1 H, H-4′-C), 3.73 (m, 1 H, Ha-5′-D), 3.65 (m, 1 H, Hb-5′-D), 3.64 (m, 1 H, Ha-5′-C), 3.51 (m, 1 H, Hb-5′-C). ^13^C NMR: δ = 157.26 (C4B), 138.90 (C3A), 138.2 (C2B), 137.68 (C6B), 129.28 (C5A), 114.06 (C3B), 112.08 (C4A), 109.64 (C5B), 97.14 (C1′-D), 93.66 (C1′-C), 86.95 (C4′-D), 84.98 (C4′-C), 76.33(C2′-D), 74.52 (C2′-C), 70.13 (C3′-D), 70.13 (C3′-C), 61.61 (C5′-C), 60.47 (C5′-D). ^15^N NMR: δ = 225.1 (N1A). Yield 5 mg (21%). Purity 90.03% (t_R_ 7.9 min). HRMS, *m*/*z*: calculated for C_18_H_25_N_4_O_6_ [M]^+^ 393.1769, found [M]^+^ 393.1472. UV λ max 289 nm.

### 2.4. Modeling of the Active Site E. coli PNP

A crystallographic model of the *E. coli* PNP [48] was previously published by Timofeev et al. and posted on the Protein Data Bank. Complete structure of protein—PDB ID code 5IU6 (Crystal structure of *E. coli* purine nucleoside phosphorylase with 7-deazahypoxanthine.

Computational docking studies were carried out on protein–ligand docking server SwissDock, based on EADock DSS [49]. The Polak–Ribiere algorithm (conjugate gradient) was used for molecular mechanics optimizations (termination conditions: root mean square (RMS) gradient of 0.1 kcal/(Å⋅mol) *in vacuo*). Analysis of molecular structures and conformational searching from related docking data using the program UCSF Chimera [50]. The structures of fleximer molecules were generated and structural optimizations were performed by using HyperChem software v8.0.6 [51]. Post-docking analysis visualized by Discovery Studio Visualizer in both 2D and 3D poses in the protein structures.

To confirm the docking result, we used a ‘pose selection’ method to re-dock 7-deazahypoxanthine with the active site (in monomer A) of *E. coli* PNP. Values of the root-mean-square deviation of atomic positions (RMSD), docking pose, accuracy, and coverage of contacts were compared with the co-crystallized structure. RMSD of atomic coordinates between two molecules was calculated using the PyMOL Molecular Graphics System, Version 2.5, Schrodinger, LLC. The obtained value has an acceptable range of RMSD docking < 1.5 Å. RMSD = 0.453 Å.

## 3. Results and Discussion

### 3.1. Enzymatic Transglycosylation of the Fleximer Heterocycle with the Formation of Isomer Products

In our previous work [46], we described the synthesis of nucleosides **15** and **18** obtained by an enzymatic transglycosylation reaction using isolated recombinant PNP and UP *E.coli*. However, we did not report this; in addition to the expected products, other nucleosides with an unknown structure were also found in the reaction mixtures (see Figure 5). We found that using molar ratios of Urd:base **12** or dUrd:base **12** exceeding 5:1 (pH 7.0) resulted in an increased content of isomeric nucleosides in the reaction mixture (chromatograms and HRMS spectra of the reaction mixtures are presented in the Appendix A). 

We increased this ratio to 10:1 and optimized the synthesis of nucleosides **15**–**19** under these conditions. According to LC-MS data, two products with one ribose residue **15** and **16** ([M+H]^+^ = 293.1233 and 293.1234) and bis-riboside product ([M]^+^ = 425.1648) were detected in the reaction mixture. In the synthesis of 2′-deoxyriboside of base **12**, the formation of two products: mono-glycosylated ([M+H]^+^ = 277.1274) and bis-glycosylated ([M]^+^ = 393.1472) products were observed.

For the isomer nucleoside synthesis, the optimal substrate ratios and the enzyme amounts in the reaction mixture were determined in order to obtain the maximum possible number of minor products **16**, **17**, and **19**.

All nucleosides were isolated from the reaction mixture using column chromatography, and their structures were determined using mass spectrometry and NMR spectroscopy. The structures of compounds **15**–**19** were confirmed by two-dimensional homo- and heteronuclear NMR spectroscopy. The NMR spectra for compounds **15** and **18** were previously published [46], and the spectra for **16**, **17,** and **19** are presented in the Appendix A.

In ribosylation of fleximer base **12**, the N1-pyridinium isomer **16** was predominantly formed in the first 24 h (Figure 6A). During the following incubation of the reaction mixture, an increase in the amount of isomer **17** and a decrease in the amount of isomer **16** were observed simultaneously. We assumed that this was due to the secondary glycosylation of isomer **16** to the N1 pyrazole. In the case of the enzymatic transfer of the deoxyribose residue, advantageous glycosylation of the pyrazole cycle **18** and formation of bis-glycosylation product **19** were detected (Figure 6B). 

We assume that if pyridinium nucleosides are formed during the transglycosylation reaction, the pH value of the reaction mixture may affect this process. We investigated the effect of pH on the conversion of fleximer base **12** into nucleosides **16**, **17,** and **19**. Due to the fact that the range of enzyme activity lies within the pH range of 5.0 to 9.0, we analyze the reaction under these conditions. The test reaction conditions were as follows: base concentration 2 mM, carbohydrate donor dUrd, or Urd concentration 20 mM. Reactions were carried out in 1.0 mL of 10 mM potassium phosphate buffer at 50 °C. To each reaction mixture, *E. coli* PNP (1.5 U) and *E. coli* UP (0.8 U) were added. The reaction was monitored by HPLC for 120 h. The results of ribosylation of base **12** at various pH values of the buffer are shown in Figure 7.

The formation of isomeric pyridine riboside **16** was primarily influenced by the pH of the reaction mixture. At pH 5–7, its conversion reached 28–44% over a period of 48 h. However, when the pH was increased to 8–9, the conversion of base **12** into riboside **16** rapidly reached 36% within 2 h, but then decreased to 10–14% over the next 48 h. At pH 8–9, the formation of the bis-ribosylation product reached a peak at 32% after 120 h. After that, the formation stopped after pyridine riboside **16** had disappeared from the reaction mixture. These findings suggest that pyridine riboside **16** undergoes secondary glycosylation at the active site of PNP. It is interesting to note that no formation of pyridine 2′-deoxyriboside was observed in the reaction mixture. The conversion of base **12** to bis-deoxyriboside **19** at pH 8–9 reached a maximum of 39% after 24 h, before decreasing. Maybe, pyridine 2′-deoxyriboside (if it is synthesized in the PNP active site) quickly converts into a bis-derivative. Another possibility is that, namely, pyrazole 2′-deoxyriboside **18** undergoes secondary glycosylation forming the bis-derivative **19**. In Figure 6B and Figure 7D, we can see an accumulation of nucleoside **18** up to 80–90% within 1 h followed by a decrease up to 60% after 40 h (Figure 7D), and then an increase back to 90% at the end of the experiment. It is most likely that pyrazole 2′-deoxyriboside **18** transforms directly into bis-2′-deoxyriboside **19** without the formation of pyridine 2′-deoxyriboside.

### 3.2. Quantum Chemical Analysis

To clarify the obtained dependence of the reaction direction on the various pH levels, a quantum chemical analysis (*ab initio*) of the fleximer **12** base was performed. This analysis modeled the structure of the base under different pH conditions (Table 1). The following parameters were calculated: total energy, partial charge, distance between atoms, and angle between the planes of the pyrazole and pyridinium cycles.

Quantum-chemical analysis of compound **12** in the free state *vs.* the form of an anion showed that the partial charge of the sp^2^ nitrogen atom of the pyridine fragments changes insignificantly, while the partial charge of the sp^2^ nitrogen atom of the pyrazole fragment changes significantly toward increasing nucleophilicity.

Based on the data presented in Table 2 and in Figure 6, we can conclude that the formation of pyrazole riboside **15** is more thermodynamically favorable than the formation of pyridine riboside **16**. 

Pyrazole riboside **15** accumulates over time, while pyridine riboside **16** does not. During the stage of substrate binding and activation, *N*_arom_ has an advantage, but the final outcome is determined by *E_T_*. It should be noted that the formation of pyridine nucleoside **16** is significantly less preferable (Δ*E_T_* = +26.2 kcal/mol) in comparison with pyrazole riboside **15** from a thermodynamic point of view (Table 2). We suppose that the reactions leading to the formation of all regioisomers are in equilibrium. The kinetics (pH) and the rates of formation of each product contribute to the process, and the overall outcome is largely determined by thermodynamic factors. 

Quantum-chemical analysis of the structure of compound **12** (*ab initio* method 6-31G**, RMS 0.3 kcal/mol) together with an analysis of literature data for structurally related heterocyclic compounds, allows us to make the following assumptions. First of all, a quantum-chemical analysis of the free molecule of base **12** has shown that the pyrazole and pyridine fragments are located in different planes (not coplanar; dihedral angle values are shown in Table 1). This orientation of two fragments of the molecule suggests a weak interaction between them. The protonation of the pyridine fragment does not significantly alter the electronic structure of the pyrazole fragment, and *vice versa*, deprotonation of the pyrazole fragment results in minor changes to the pyridine fragment (Table 1).

Analysis of two pyrazole structures of compound **12** showed at pH 7 and 8–9, as expected, close symmetry and similarity of electronic structure. Based on these data, one could expect a low nucleophilicity of both pyrazole sp^2^ hybridized nitrogen atoms (the partial charges of N^1^ = −0.4221 *e*, N^2^ = −0.3061, and *N*^1(2)^ ≈ −0.4140 *e* at pH 7 and 9, respectively), on the one hand, and a high nucleophilicity and affinity for a proton of the pyridine nitrogen atom (*N*_arom_ ≈ −0.5759 *e*; sp^2^ hybridized atom) and amino group (NH_2_ ≈ −0.7620 *e*), on the other. The observed change in the rate of formation of mono- and bis-ribosides **15** and **17** agrees satisfactorily with an increase in the nucleophilicity of the pyrazole sp^2^ nitrogen atom (*vide supra*).

Quantum-chemical analysis of compound **12** in its free form, depending on the anion form, showed that the partial charge of the sp^2^ nitrogen atom in the pyridine fragment changed insignificantly: [Δ*N*_arom_ = −0.5803 − (−0.5759) = −0.0044 *e*], while the partial charge of the sp^2^ nitrogen atom of the pyrazole fragment changes considerably [Δe = −0.4140 − (−0.3061) = −0.1079 *e*] toward an increase in nucleophilicity. It is likely that this is a significant factor in the rate of riboside regioisomer synthesis at the initial stage of the reaction when the pH of the reaction buffer is between 8 and 9.

The reaction of ribosylation and 2-deoxyribosylation catalyzed by *E. coli* PNP at various pH values of the reaction mixture was studied in order to shed light on the possible mechanism of binding and activation of heterocyclic substrates in the catalytic site of the enzyme. In addition, attention was directed to elucidating the role of thermodynamic parameters of key intermediate structures in the formation of the final reaction products.

When the pH of the reaction mixture changes from 5–6 (most likely protonation of the amino group) to 8–9 (most likely deprotonation of the pyrazole fragment) (Table 1), resulting in (1) the enhancement of the nucleofilicity of pyrazole nitrogen atoms (from *N*^2^ ≈ −0.2590 *e* to *N*^1(2)^ ≈ −0.4140 *e*), and (2) the rate of formation of primary riboside **15** and secondary bis-riboside **17** gradually increases, and after 48 h mixtures of these compounds are formed in a ratio of 35%/2% (pH 5–6) and 65%/13% (pH 8–9), respectively. The conversion of **16** to bis-riboside **17** proceeded synchronously with the formation of the former and reached 13% at pH 8–9 of the reaction medium. Based on these data, it can be assumed that the pyridine riboside **16** is initially formed, which is then ribosylated to bis-riboside **17**.

In the case of the formation of a pyridine nucleoside **16** catalyzed by *E. coli* PNP, which is non-precedent to our knowledge, the hydrogen bond between the amino group and Asp204 seems to play a key role. It should be noted that, as far as we know, the enzymatic synthesis of pyridine nucleosides has not yet been described. Earlier, Shugar et al. described the phosphorolysis of nicotinamide riboside by PNP from calf spleen, the reaction was not reversible [52]. It is known that the overwhelming majority of reactions catalyzed by nucleoside phosphorylases are reversible and the equilibrium shifts toward the thermodynamically most stable product. Appendix A shows the equilibrium state of all the components involved in the base **12** ribosylation process (see Appendix A). 

Trans-2-deoxyribosylation of **12** proceeded more efficiently compared to the ribosylation, and the formation of the nucleoside **18** reached 70–80% and ca. 50% in 2 h at pH 6–7 and 5, 8, and 9, respectively, remaining without essential changes for the next 48 h. The formation of bis-2′-deoxyriboside **19** during the first 24 h of the reaction time reaches 15% at pH 6 and 45% at pH 6–8 and then slowly decreases to 10 and 35%, respectively; at pH 5, nucleoside **19** did not appear in the reaction medium. The diagrams of the formation of nucleosides **15** and **18** at different pH values under standard reaction conditions are somewhat different. These differences are apparently associated with a higher substrate activity of 2-deoxyribosyl phosphate (dRib-1P) *vs.* ribosyl phosphate (Rib-1P), and on the other hand, with the known lower chemical stability and higher substrate activity of 2′-deoxyglycosides *vs.* comparison with ribosides. At pH 5–6, the amino group is protonated first (pKa ca. 9), then the ***N***_arom_ atom and, finally, one of the pyrazole nitrogen atoms. In the case of the synthesis of pyrazole glycosides mediated by *E. coli* PNP, one can consider the binding and activation of the substrate by the formation of a hydrogen bond (1) Asp204 with an amino group or sp^2^ hybridized nitrogen atom and (2) Ser90 with the sp^3^-hybridized pyrazole nitrogen atom [45,48].

The formation of pyrazole glycoside and pyridine glycoside occurs through two different mechanisms. In the first case, the participation of *E. coli* PNP Ser90 in substrate binding and activation is similar to that previously assumed by us in the enzymatic transglycosylation of 8-aza-7-deazapurines [45,48]. Comparative analysis of the structures of adenosine and pyrazole riboside suggests the participation of the pyridine nitrogen atom in the conversion of the heterobase into the nucleoside.

### 3.3. Structure Determination of Isomeric Nucleosides: Comparison of NMR Data of a Pyridinium Derivative with Known Compounds

For all minor compounds **16**, **17**, and **19** in the NMR spectra, a singlet signal could be expected for proton H2, and a doublet signal for proton H6, and a doublet and a doublet of doublets were observed in the real spectrum, respectively (Figure 8 and Appendix A). In addition, there was a significant difference in the chemical shifts of the NH2 protons: 7.42 and 8.55 ppm (all signals were wide). For protons of the amino group, we should have expected closer values of chemical shifts due to their equivalence. This led us to compare the spectrum of the obtained compound with the spectra of the most similar structures described in the literature (Figure 9). 

We attempted to compare the NMR spectra data of nucleoside **16** with those of a structurally similar compound—nicotinamidriboside [39]. The shapes of the signals and the chemical shifts of the aromatic protons H2 and H6 were not correlated with those of the aromatic protons H2 and H6 of nicotinamidriboside, but they are very similar to the analog atoms of clitidine, the structure of which was previously established by NMR and RSA [53].

Pyridinium nucleoside **16** can exist in two tautomers: the amino form and the imino form. (Figure 8). At first, we thought that the product might be in the amino form. However, the literature presents NMR data for clitidine, with chemical shift values for the imino group protons of 7.91 and 9.02 ppm [53]. For nucleoside **16**, these values were 7.42 and 8.55 ppm. The chemical shifts and multiplicities of the H2 signals for clitidine and nucleoside **16** were 8.72 ppm (d) and 8.55 ppm (d), respectively. The H6 signal for clitidine was 8.11 ppm (dd) and that for nucleoside **16**—8.35 ppm (dd).

Thus, it is clear that the spectral signals of our nucleoside **16** and clitidine were similar. This may indicate the structural similarity between these two compounds. It can be concluded that nucleosides **16**, **17**, and **19** exist in the imino form (Figure 9). 

### 3.4. Modeling the Interaction of Fleximers in the Active Site of the E. coli PNP

To explain the formation of minor products in the glycosylation reaction of base **12**, we attempted to predict the interaction between the ligand (fleximer nucleosides in this case) and the PNP active site of *E. coli* using a molecular modeling method. Before docking with fleximer, the crystallographic structures of *E. coli* purine nucleoside phosphorylase, posted on the Protein Data Base website were analyzed. This was a necessary step in order to determine the most common interactions (hydrogen bond formation, π–π stacking, etc.) between the active site and various ligand variants. The PNP complexes were studied with the following ligands in the active site: 7-deaza-hypoxanthine (PDB: 1PR5) [48], acyclovir (PDB: 5I3C) [54], and formicin A (PDB: 1K9S) [12].

The purine base interaction site is formed by the following residues: Ser90, Cys91, Gly92, Ala156, Phe159, Phe167, Val178, Glu179, Met180, Asp204, and Ile206. These four residues, Ala156, Phe159, Val178, and Met180, combine to form a hydrophobic domain that surrounds the purine base. Asp204 interacts with both the N7 atom and the substituent on the C6 atom of the purine base, while Phe159 forms a 60-degree angle with the plane of the purine cycle, providing a π–π interaction between the aromatic systems and keeping them close together. The ribose interaction site is composed of Met64, Phe159, Val178, Glu179, Met180, and Glu181 residues. Glu181 forms hydrogen bonds with the 2′-and 3′-hydroxyl groups of the ribose [16].

Docking of the natural substrate (inosine in this case) into the active site was carried out to determine its optimal orientation. This process was achieved by searching for suitable conformations (see Figure 10).

After successfully determining the optimal position of the natural substrate, the new fleximer nucleosides were docked. The main objective of this step was to find positions analogous to those of the original substrate and to examine the interactions between the novel ligands and the amino acid residues in the active site. 

A variety of interaction types, including hydrogen bonds, van der Waals forces, and ionic interactions, as well as other factors, were taken into account during the analysis. The docking process was carried out in a 10-angstrom cube, providing sufficient space for the ligand to fit within the catalytic site. Based on the calculated binding energy values, the complexes were ranked in order of lowest energy. The complexes with the lowest energy, which also exhibited specific conserved ligand–protein interactions, were chosen for further analysis. 

The same amino acid residues are responsible for the binding of 4-(4-aminopyridine-3-yl)-1H-pyrazole **12** and its orientation at the binding site of the enzyme, as well as for its interaction with a natural purine base. The purine base occupies a region formed by Ser90, Met180, Ser203, Asp204, and the hydrophobic Phe159. 

Due to the flexible structure provided by the C-C bond between the pyrazole and aminopyridine cycles, the base can occupy different positions in the active site. Special attention is paid to the interactions between the aromatic cycles of the aminopyridine and pyrazole parts and the Phe159 amino acid residue through π–π interactions. (Figure 11A).

In addition, the formation of a hydrogen bond with Asp204 is also an important factor. When the pyridine cycle is placed close to residue Phe159, and a hydrogen bond forms with the amino group of Asp204, it enables the nitrogen of the pyrazole cycle to be in a specific (normal, as in natural purine) position in the active site to interact with the ribose phosphate and form a glycosidic bond (Figure 11B).

In contrast, when the pyrazole cycle is located directly next to Phe159, and a hydrogen bond forms between the proton and Asp204, and, as mentioned earlier, the nitrogen of the pyridine cycle is sufficiently nucleophilic, glycosylation occurs (Figure 12A,B).

As for pyridinium riboside **16**, there are two possible locations for it in the active site (see Figure 13). The presence of a relatively large space in the area of amino acid residues Phe159, Val178, and Asp204 allows the ribose residue to fit into that space, and the heterocyclic base can occupy a position that is suitable for subsequent glycosylation. Perhaps it is this “upside down” arrangement that promotes secondary glycosylation and the formation of minor bis-riboside **17**. This assumption is supported by data from pH experiments, which show that, initially, the formation of pyridinium ribose predominates in the reaction mixture, and then, bis-glycoside slowly accumulates while the amount of pyridinium ribose decreases to zero.

Docking experiments have shown that the fleximer bases and ribosides are located in the selected binding region. The binding free energy (ΔG) and the amino acid residues that interact with the molecules are listed in Table 3.

## 4. Conclusions

The glycosylation of 4-(4-aminopyridine-3-yl)-1H-pyrazole fleximer base using recombinant *E. coli* PNP resulted in the formation of not only the expected N1-pyrazole nucleoside, but also two previously unreported ribosides: 4-imino-pyridinium and a bis-ribose derivative of N1-pyridine and N1-pyrazole. The structures of compounds were determined using NMR spectroscopy and mass spectrometry. The formation of the pyridinium riboside occurred predominantly at the beginning of the reaction and was strongly dependent on the pH. Subsequent to this, an increase in the amount of the bis-ribose isomer and a decrease in that of pyridinium riboside were observed, which were caused by secondary glycosylation of the nitrogen atom of the pyrazole cycle. In the case of 2′-deoxyribosylation, glycosylation along the pyrazole cycle prevailed and only a bis-glycosylation product was formed. To better understand the mechanism of *E. coli* PNP functioning, the analysis of the enzyme active site and the possible structures of the regioisomeric complexes and bis-glycosylation products in the active site was carried out using computer modeling methods *in silico*.

## Figures and Tables

**Figure 1 biomolecules-14-00798-f001:**
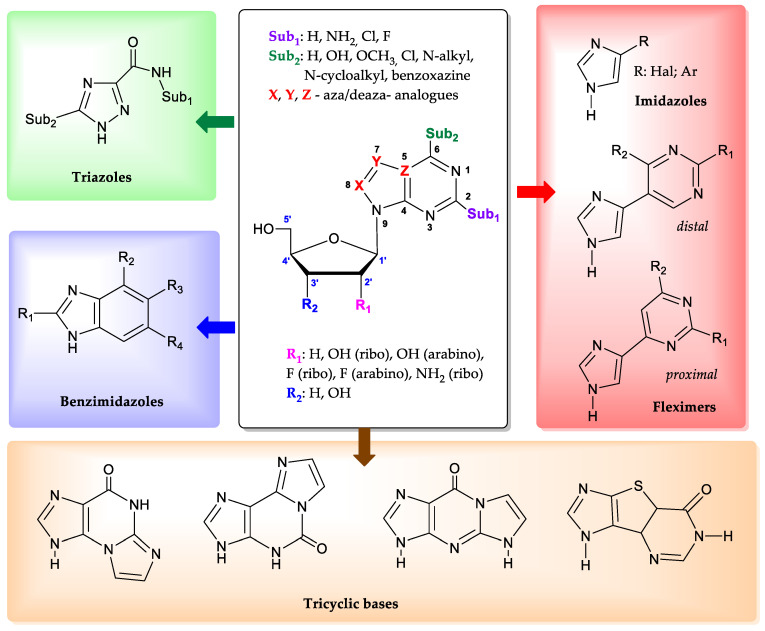
The structures of modified bases and nucleosides that *E. coli* PNP accepts as substrates.

**Figure 2 biomolecules-14-00798-f002:**
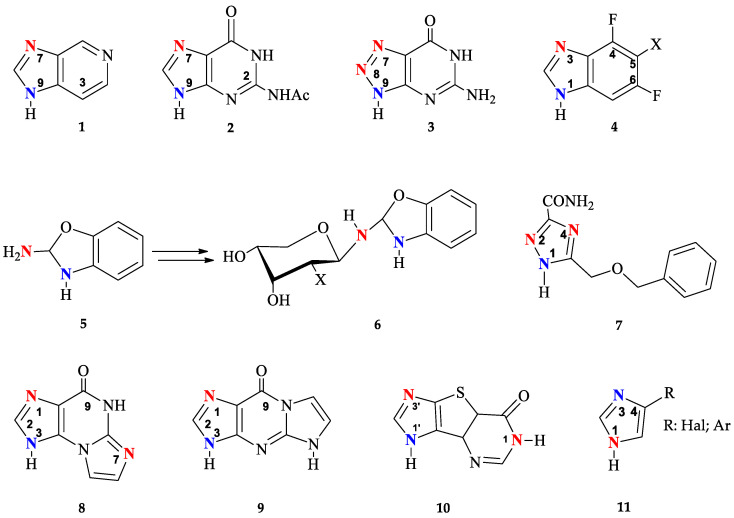
Modified heterocyclic bases with different glycosylation sites (highlighted in blue—normal and red—unusual).

**Figure 3 biomolecules-14-00798-f003:**
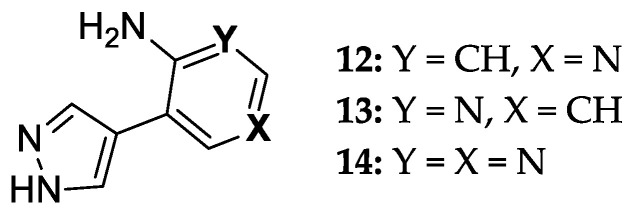
8-Aza-7-deazapurine fleximer bases.

**Figure 4 biomolecules-14-00798-f004:**
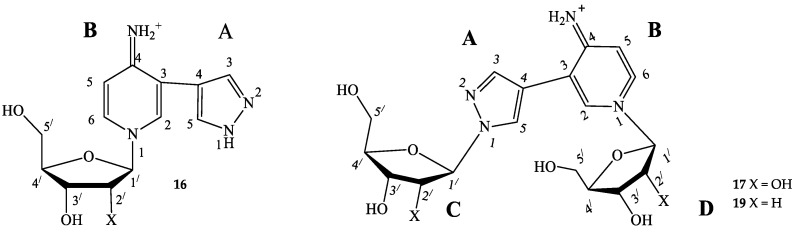
Structural formulas with the numbering of atoms in cycles. A—pyrazole cycle, B—pyridine cycle, C—the first ribose/2-deoxyribose residue, D—the second one.

**Figure 5 biomolecules-14-00798-f005:**
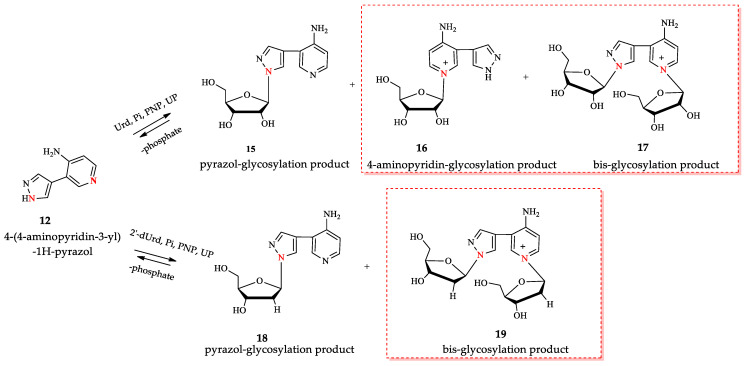
Enzymatic transglycosylation of fleximer base **12** with the formation of isomer products (highlighted with a red border). The nitrogen atoms involved in the glycosylation reaction are shown in red.

**Figure 6 biomolecules-14-00798-f006:**
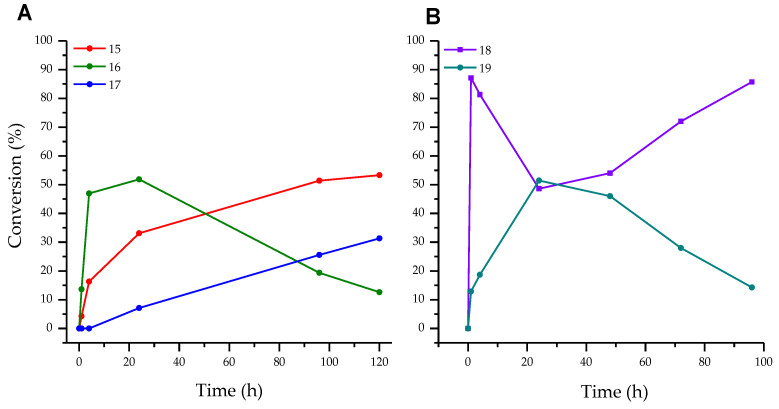
(**A**)—Conversion of base **12** to ribosides **15**, **16,** and bis-riboside **17**, (**B**)—Conversion of base **12** to 2′-deoxyriboside **18** and bis-2′-deoxyriboside **19**.

**Figure 7 biomolecules-14-00798-f007:**
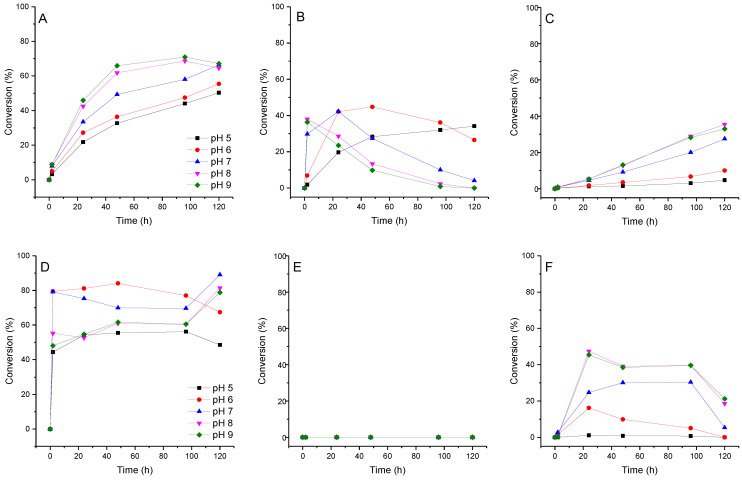
Conversion of fleximer base **12** to pyrazole riboside **15**, aminopyridinium riboside **16**, and bis-riboside **17** at various pH. (**A**)—**15** (pyrazole riboside), (**B**)—**16** (pyridine riboside), (**C**)—**17** (bis-riboside), (**D**)—**18** (pyrazole 2′-deoxyriboside), (**E**)—pyridine 2′-deoxynucleoside was not found in the reaction, (**F**)—**19** (bis-2′-deoxyriboside).

**Figure 8 biomolecules-14-00798-f008:**
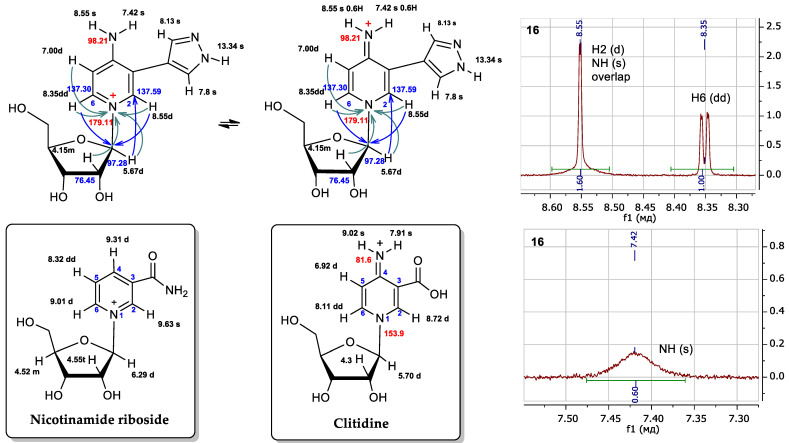
Possible structures of fleximer riboside **16** and comparison of proton chemical shifts in the NMR spectra of nicotinamidriboside [39] and clitidine [53]. Fragments ^1^H NMR spectrum of nucleoside **16**. The ^1^H chemical shifts are shown in black, the ^13^C in blue and the ^15^N in red. Turquoise arrows depict the interactions between the protons and the nitrogen atom of pyridine cycle, and blue arrows are the interactions of the protons and the first carbon atom of ribose.

**Figure 9 biomolecules-14-00798-f009:**
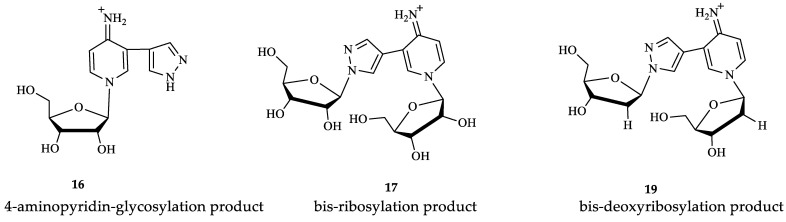
Structures of minor fleximer nucleosides.

**Figure 10 biomolecules-14-00798-f010:**
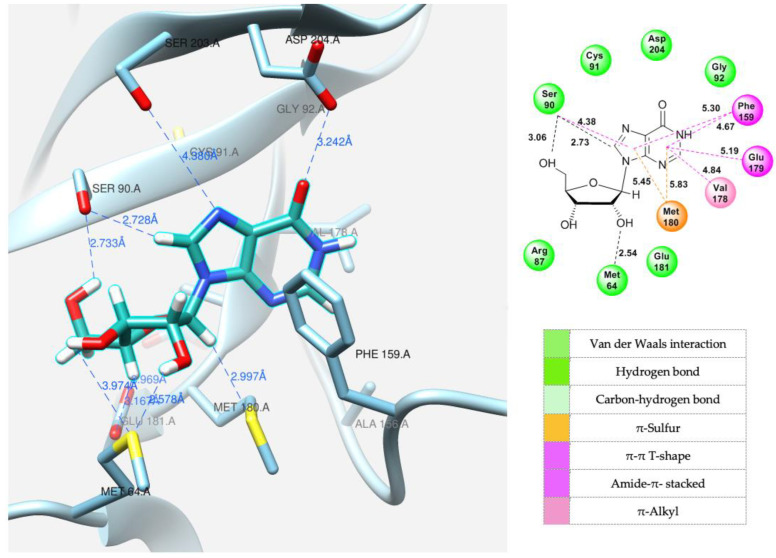
Model of inosine interaction in the PNP active site.

**Figure 11 biomolecules-14-00798-f011:**
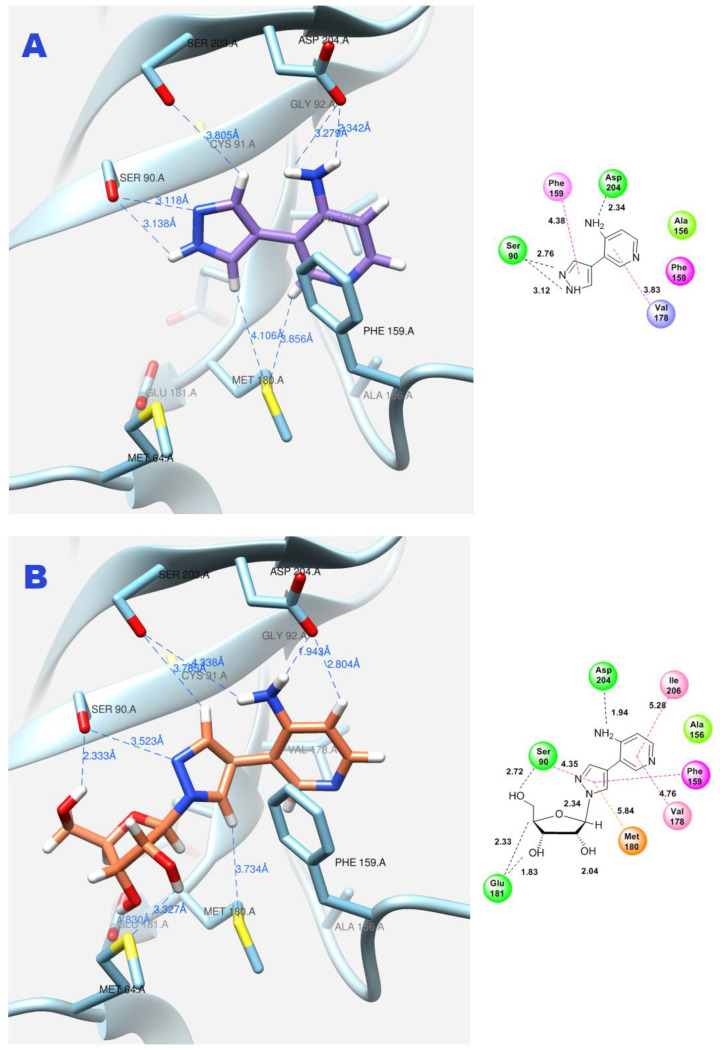
Interactions between fleximer base 12 (**A**) and riboside 15 (**B**) in the PNP active site.

**Figure 12 biomolecules-14-00798-f012:**
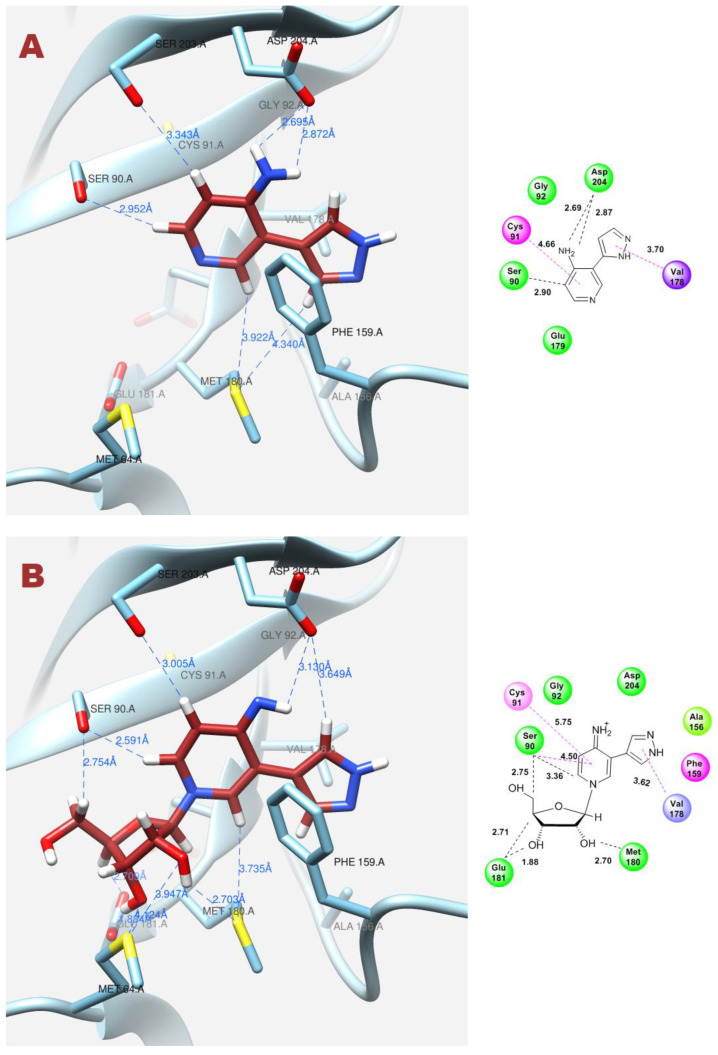
Interactions between fleximer base **12** (**A**) and riboside **16** (**B**) in the PNP active site.

**Figure 13 biomolecules-14-00798-f013:**
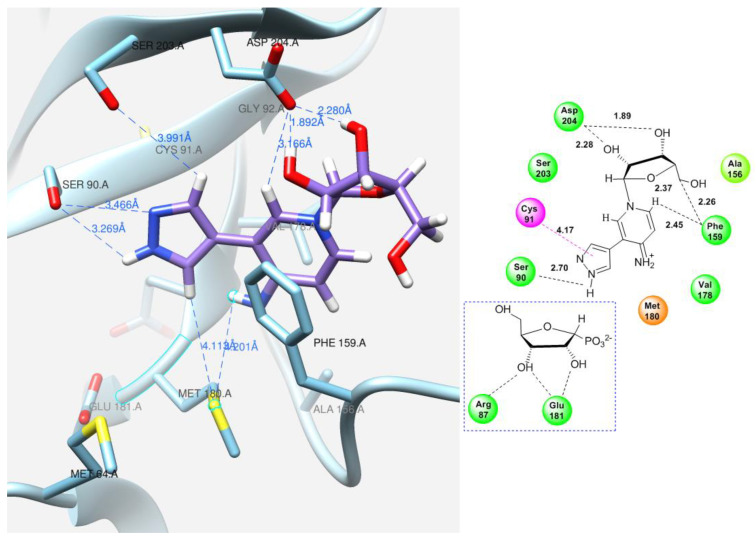
Interactions of the fleximer riboside **16** in the PNP active site, “alternative position”.

**Table 1 biomolecules-14-00798-t001:** Quantum Chemical Analysis of base **12** at different pH values by PM3 and an *ab initio* amber FF 6-31G** methods. The calculation is carried out according to the PM3 method and then recalculated using the *ab initio* amber FF 6-31G** method, Py—4-aminopyridyl.

pH 5–6	pH 7	pH 8–9
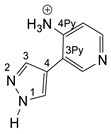	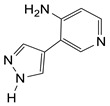	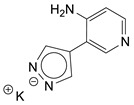
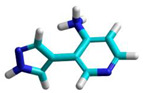	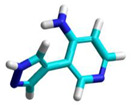	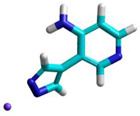
*E_T_* = −329,907.8 kcal/mol	*E_T_* = −329,688.9 kcal/mol	*E_T_* = −430,902.4 kcal/mol;
N^1^ = −0.4000 *e* N^2^ = −0.2590 *e*	N^1^ = −0.4221 *e* N^2^ = −0.3061 *e*	N^1^ = −0.4150 *e* N^2^ = −0.4130 *e*
NH_3_ = −0.6600 *e*	NH_2_ = −0.7620 *e*	NH_2_ = −0.7610 *e*
N_arom Py_ = −0.5010 *e*	N_arom Py_ = −0.5759 *e*	N_arom Py_ = −0.5803 *e*
*-*	*-*	*-*
C^3^-C^4^-C^3Py^-C^4Py^ −61.7°	C^3^-C^4^-C^3Py^-C^4Py^ −51.0°	C^3^-C^4^-C^3Py^-C^4Py^ −46.2°
*-*	*-*	*-*
N^1^<—>NH_2_ 4.302 Å	N^1^<—>NH_2_ 4.830 Å	N^1^<—>NH_2_ 4.828 Å
N^2^<—>N_ar/para_ 5.714 Å	N^2^<—>N_ar/para_ 5.690 Å	N^2^<—>NH_2_ 4.316 Å
	-	N^1^<—>N^arom^ 5.676 Å
	-	N^2^<—>N^arom^ 5.921 Å

**Table 2 biomolecules-14-00798-t002:** Quantum-Chemical Analysis of nucleosides **15** and **16** by PM3 and an *ab initio* amber FF 6-31G** methods.

Pyrazole Riboside 15	Pyridine Riboside 16
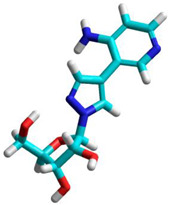	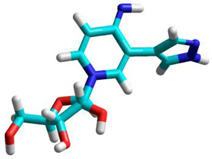
*E*_T_ = −639,329.0 kcal/mol	*E*_T_ = −639,302.8 kcal/mol
Δ*E*_T_ = −26.2 kcal/mol	Δ*E*_T_ = +26.2 kcal/mol
C^3′-^exo; DM 4.316 Debye’s	C^2′-^endo/C^1′-^exo (twist); DM 8.833 Debye’s
*N*_arom/para_ = −0.5757 *e*	*N*^1^ C^1′^Rib = −0.7586 *e*; *N*H = −0.6792 *e*;
*N*^2^<—>*N*H_2_ 4.2685Å	*N*H = −0.4229 *e*; *N* = −0.3087 *e*
*N*^2^<—> *N*_Py_ 5.9227 Å	O^4′^-C^1′^-*N*-C^1″^H +44°
O^4′^-C^1′^-*N*^1^-*N*^2^ +68.8°	O^4′^-C^1′^-*N*-C^6″^H −142°
C^3^-C^4^-C^3Py^-C^4Py^ −48.6°	*N*^3^H-C^4^-C^3″^-C^4″^H*N*H −115°
*-*	*N*^3^H-C^4^-C^3″^-C^2″^H*N* +62°

**Table 3 biomolecules-14-00798-t003:** Docking results of inosine, fleximer base 12 and ribosides 15 and 16.

Model		Cluster	Energy (Kcal/mol)	ΔG (Kcal/mol)	Contacting Amino Acid Residues with Interaction Type	
					Hydrogen Bond	π–π Interaction
Figure 10	Inosine	7	13.56	−6.82	Ser90, Met64	Met180, Phe159, Val178, Glu179
Figure 11A	Fleximer base **12**	3	3.76	−6.55	Ser90, Asp204	Cys91, Val178
Figure 11B	Pyrazole riboside **15**	3	63.71	−7.07	Ser90, Cys91, Glu181, Asp204	Phe159, Val179, Met 180, Ile206
Figure 12A	Fleximer base **12**	4	7.00	−6.07	Ser90, Asp204	Cys91, Val178
Figure 12B	Pyridinium riboside **16**	6	12.99	−7.80	Ser90, Met180, Glu181	Cys91, Val178
Figure 13	Pyridinium riboside **16**	0	25.75	−7.60	Ser90, Phe159, Asp204	Cys91

## Data Availability

The data presented in this study are available within the article.

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
