# Peer review of "Enzymatic Transglycosylation Features in Synthesis of 8-Aza-7-Deazapurine Fleximer Nucleosides by Recombinant E. coli PNP: Synthesis and Structure Determination of Minor Products"

_biomolecules, 2024, doi:10.3390/biom14070798_

Round 1
Reviewer 1 Report
Comments and Suggestions for Authors
This is a well written manuscript describing the use of E. coli PNP to synthesize fleximer nucleosides.
Author Response
Thank you for your appreciation of our research.
Reviewer 2 Report
Comments and Suggestions for Authors
This paper describes the enzymatic synthesis of ribo- and 2’-deoxyribonucleoside analogues bearing fleximer heterocycles as nucleobases. Fleximer nucleoside analogues have been proposed as molecular probes for biomolecular studies, but also as potential antiviral and antitumor agents. The Authors have performed an in-depth investigation of biotransformations catalyzed by the purine nucleoside phosphorylase (PNP) from E. coli, focusing on the formation of minor, “atypical” reaction by-products, resulting from non-regioselective glycosylation of the modified nucleobase, and trying to elucidate the mechanism behind their formation, thereof. This investigation was supported by a molecular modeling study on the mechanism of binding and activation of heterocyclic substrates in the enzyme active site.
This article is in continuation with the previous research of some of the Authors on the enzymatic synthesis of nucleoside analogues catalyzed by nucleoside phosphorylases. The current study contributes to further enrich the knowledge about the reactivity of this enzyme family toward modified nucleobases and nucleosides. In this frame, the solid experience and background accumulated over the years by some of the Authors in nucleoside chemistry clearly emerges in the Introduction.
The aim is clear. Results and Discussion are overall solid and well-organized. Nevertheless, some criticisms raised upon the reviewing process that Authors should consider both in the text and in the rebuttal letter.
1) The Authors refer indistinctly to glycosylation and/or transglycosylation (even in the title). However, these reactions are mechanistically (and in practice) very different. As far as it is reported in the Experimental section, the reactions are two-enzyme transglycosylations and not one-enzyme glycosylation (see, in particular, the paragraph 2.2). They used uridine phosphorylase (UP) coupled to PNP to perform the synthesis of the nucleosides, by exploiting Urd or 2’-dUrd as sugar donor. This is the major issue that the Authors should explain and address in the paper that, otherwise, results to be confusing.
2) In the Experimental Section (General) information was not provided about neither the source of enzymes (supplier? Were the biocatalysts produced “in house”? How?) nor the enzymatic assay to test their specific activity. This lack of information somehow reflects in the lack of consistency in reporting the activity of the used enzymes. The measure unit (U) should be used consistently throughout the text for “enzymatic unit”. The definition of enzymatic unit (U) should be added in the Experimental Section-General
3) About the artwork (graphics), I warmly suggest to report the results by using the same scale in each graph. This would result in a more straightforward comparison of data. For instance, Fig. 6A and 6B: the axis of Fig. 6A has 50% as the highest value, whereas the axis of Fig. 6 B has 100% as the highest value. It would be clearer, at a glance, to have both graphs reporting 100% conversion as the highest value in order to better appreciate the different trend of the riboside and 2’-deoxyriboside formation, respectively. Same as for Fig. 7 A-F
4) The legend of Figure 1 must be modified (Figure 1. PNP E. coli substrate structures). The current caption is not clear. An option could be: Substrates recognized by the PNP from E. coli.
5) Line 124: nucleoside products instead of nucleotide products
6) Experimental Section, General: the country of the suppliers should be reported
7) Experimental Section, Paragraph 2.2: the Authors should report how the reaction conversion was calculated
8) References style must be double-checked for consistency
Comments on the Quality of English LanguageNone. See the general comment
Author Response
Comments 1:The Authors refer indistinctly to glycosylation and/or transglycosylation (even in the title). However, these reactions are mechanistically (and in practice) very different. As far as it is reported in the Experimental section, the reactions are two-enzyme transglycosylations and not one-enzyme glycosylation (see, in particular, the paragraph 2.2). They used uridine phosphorylase (UP) coupled to PNP to perform the synthesis of the nucleosides, by exploiting Urd or 2’-dUrd as sugar donor. This is the major issue that the Authors should explain and address in the paper that, otherwise, results to be confusing.
Response 1: The incorrect term "glycosylation" has been corrected to "transglycosylation" in the text of the manuscript. We also added a description: «A two-enzyme approach using E. coli UP and PNP was used to perform the transglycosylation reaction of fleximer bases. Uridine phosphorylase (E. coli UP) was used to generate ribose-1-phosphate and 2′-deoxyribose- 1-phosphate from Urd and dUrd, respectively».
Comments 2: In the Experimental Section (General) information was not provided about neither the source of enzymes (supplier? Were the biocatalysts produced “in house”? How?) nor the enzymatic assay to test their specific activity. This lack of information somehow reflects in the lack of consistency in reporting the activity of the used enzymes. The measure unit (U) should be used consistently throughout the text for “enzymatic unit”. The definition of enzymatic unit (U) should be added in the Experimental Section-General.
Response 2: The following paragraph has been added to the experimental section: «Plasmid vectors containing the genes encoding EcoPNP and EcoUP constructed in previous research were used for the transformation of competent E. coli ER2566 (New England Biolabs) strain cells [34]. The resulting producer strains were cultivated at 37°C in the Luria–Bertani medium (10 g. tryptone, 5 g. yeast extract 10 g. NaCl per 1L) with 100 µg/mL ampicillin. Upon reaching the optical density of A595=0.8, the cell cultures were supplemented with 0.4 mM of IPTG, followed by further cultivation of 4 h at 37◦C, which allowed to obtain enzymes in soluble form.
The cell biomass was harvested through centrifugation and resuspended (1:10 w/v) in buffer containing 50 mM Tris-HCl pH 8.0, 5 mM EDTA, 1 mM benzydamine hydrochloride and 1 mM PMSF. The cell suspension was subjected to ultrasonic disintegration and centrifugation to remove the cell debris. The resulting supernatant was applied onto a column packed with Q Sepharose FF sorbent (Cytiva, Washington, D.C., U.S), equilibrated with buffer containing 50 mM Tris-HCL pH 8.0, 5 mM EDTA. The proteins were eluted with a linear gradient 0-500 mM NaCl. The fractions containing the desired enzymes were pooled, concentrated on the YM-30 membrane (Millipore) and applied onto columns packed with Superdex 200 (GE Healthcare, Chicago, IL, United States) sorbent, pre equilibrated the final buffer: 50 mM Tris-HCl pH 8.0, 200 mM NaCl. 0.04% NaN3. Fractions containing the target enzymes were pooled, concentrated on the YM-30 membrane (Merck Millipore, Burlington, MA, United States) to a concentration of approximately 30 mg/ml. The resulting aliquots of the enzyme’s solution were stored at -80°С for further experiments. We have measured the protein concentration according to Bradford and determined the enzyme purity through SDS PAGE. Solutions of recombinant E. coli UP and PNP in 10 mM potassium phosphate buffer (pH 7.0) with activities 1700 and 1400 units per mL, respectively, were used»
Comments 3: About the artwork (graphics), I warmly suggest to report the results by using the same scale in each graph. This would result in a more straightforward comparison of data. For instance, Fig. 6A and 6B: the axis of Fig. 6A has 50% as the highest value, whereas the axis of Fig. 6 B has 100% as the highest value. It would be clearer, at a glance, to have both graphs reporting 100% conversion as the highest value in order to better appreciate the different trend of the riboside and 2’-deoxyriboside formation, respectively. Same as for Fig. 7 A-F
Response 3: Corrected.
Comments 4: The legend of Figure 1 must be modified (Figure 1. PNP E. coli substrate structures). The current caption is not clear. An option could be: Substrates recognized by the PNP from E. coli.
Response 4: Corrected. « The structures of modified bases and nucleosides that E. coli PNP accepts as substrates.»
Comments 5: Line 124: nucleoside products instead of nucleotide products.
Response 5: Corrected.
Comments 6: Experimental Section, General: the country of the suppliers should be reported
Response 6: The following paragraph has been added to the experimental section: «Conversion was calculated using HPLC data as the ratio of the peak area of the product to the sum of the peak areas of the product and the starting base.»
Comments 7: Experimental Section, Paragraph 2.2: the Authors should report how the reaction conversion was calculated
Response 7: Corrected.
Comments 8: References style must be double-checked for consistency
Response 8: The references style has been corrected.
Reviewer 3 Report
Comments and Suggestions for Authors
The paper by Eletskaya et al. describes some interesting data on transglycosylation. In general, it is suitable to be published in 'Biomolecules'; however, it suffers with a number of problems, which have to be corrected prior to publication (including some minor corrections/typos required):
1. The 'Introduction' section looks like a short review of the authors' contribution to the field of PNP transglycosylation. It should be significantly confined to data related to the topic of the ability of PNP to catalyze the formation of imidazole/pyridine nucleoside analogs. In particular, lines 67-69 and 78-89 seem to be needless. Self-citation should be significantly limited. Eg., instead of numerous refs 11-19, at least two representative examples would be definitively sufficient.
2. In contrast, the beginning of Section 3 lacks any refs, which should be present there.
3. In Section 3, it is unclear what the previous contribution is and where the current one starts.
4. Lack of consistency: Conditions for preparation of cpds. 15-19 are doubled in lines 235-236 and 247-250.
5. Figure 5: what do the colors of the frames mean? The meaning should be explained in the caption.
6. Line 258: use hours instead of days.
7. Line 261: 'isomer 15 to the N1 pyrazole' should be 'isomer 16 to the N1 pyrazole', probably.
8. Line 263+: Figure 6B: More discussion is required since the reaction mixture composition is really intriguing. Why did an opposite reaction occur after 24 h? What was the composition of the r.m. between 24 and 96 h? The data areincomplete.
9. Figure 7 caption: terms riboside/glycoside are used interchangeably; this may be misleading.
10. Paragraph 282-289: the interpretation of the results is dramatically too scarce.
11. Paragraph 304-311: discuss it in reference to Fig. 6.
12. Lines 316-318: 'First of all, a quantum-chemical analysis of the free molecule of base 12 has shown that the pyrazole and pyridine fragments are located in different planes (not coplanar)': This statement is not supported with data. Table 2 and Fig. 8 do not show the geometry of 12.
13. Fig. 8: What do the red/green colors mean?
14. Lines 326-327: 'two N1(2)-pyrazole tautomeric structures of compound 12': at pH 5-6? – it needs clarification, Which are N1 and N2? The structures should be numbered and referenced in the text.
15. Paragraphs 326-357: the charges in the text are partially inconsistent with Fig. 8 (eg. -0.579, -0.5803, -0.5759) or Table 1 (-0.307). 'NH2 -0.762 e' is not shown in Fig. 8.
16. Paragraph starting in line391: "In the NMR spectra of pyridinium nucleosides…' – the spectra are required to be shown here. At least 16, 17, and 19, with the discussed signals highlighted.
17. L. 399, 411, 415: '16' should be in bold
18. Fig. 9: some "H" symbols are shown, some are omitted – and look like methyl groups.
19. Fig. 9: why are some H's highlighted in green? The meaning of this should be explained in the caption.
20. L. 482-483: something is wrong: 'Ошибка! Источник ссылки не найден'
The language is fully understandable; however, a native speaker proofreading would be advantageous.
Author Response
Comments 1: The 'Introduction' section looks like a short review of the authors' contribution to the field of PNP transglycosylation. It should be significantly confined to data related to the topic of the ability of PNP to catalyze the formation of imidazole/pyridine nucleoside analogs. In particular, lines 67-69 and 78-89 seem to be needless. Self-citation should be significantly limited. Eg., instead of numerous refs 11-19, at least two representative examples would be definitively sufficient.
Response 1: We would like to keep the introduction and references to 11-19, as it provides all currently known cases of non-specific transglycosylation reactions. This information may be of interest to a wide range of readers.
Comments 2: In contrast, the beginning of Section 3 lacks any refs, which should be present there.
Response 2: Corrected.
Comments 3: In Section 3, it is unclear what the previous contribution is and where the current one starts.
Response 3: Corrected. « In our previous work [33], we described the synthesis of nucleosides 15 and 18 obtained by an enzymatic transglycosylation reaction using isolated recombinant PNP and UP E.coli. However, we did not reported that, in addition to the expected products, other nucleosides with an unknown structure were also found in the reaction mixtures»
Comments 4: Lack of consistency: Conditions for preparation of cpds. 15-19 are doubled in lines 235-236 and 247-250.
Response 4: Duplicate information has been removed.
Comments 5: Figure 5: what do the colors of the frames mean? The meaning should be explained in the caption.
Response 5: Corrected. «Enzymatic transglycosylation of fleximer base 12 with the formation of isomer products (highlighted with a red border).»
Comments 6: Line 258: use hours instead of days.
Response 6: Corrected.
Comments 7: Line 261: 'isomer 15 to the N1 pyrazole' should be 'isomer 16 to the N1 pyrazole', probably
Response 7: Corrected.
Comments 8: Line 263+: Figure 6B: More discussion is required since the reaction mixture composition is really intriguing. Why did an opposite reaction occur after 24 h? What was the composition of the r.m. between 24 and 96 h? The data are in complete.
Response 8: Unfortunately, no reaction monitoring was done for this experiment between 24 and 96 hours. This was the first experiment where non-specific products of enzymatic glycosylation were detected. But Figure 7 shows more detailed data (2, 24, 48, 96 and 120h)
Comments 9: Figure 7 caption: terms riboside/glycoside are used interchangeably; this may be misleading.
Response 9: Corrected. «Figure 7. Conversion of fleximer base 12 to pyrazole riboside 15, aminopyridinium riboside 16 and bis-riboside 17 at various pH. A – 15 (pyrazole riboside), B – 16 (pyridine riboside), С – 17 (bis-riboside), D – 18 (pyrazole 2’-deoxyriboside), E – pyridine 2’-deoxynucleoside was not found in the reaction, F – 19 (bis- 2’-deoxyriboside).»
Comments 10: Paragraph 282-289: the interpretation of the results is dramatically too scarce.
Response 10: Corrected.
Comments 11: Paragraph 304-311: discuss it in reference to Fig. 6.
Response 11: The reference to Fig 6 is included into the discussion.
Comments 12: Lines 316-318: 'First of all, a quantum-chemical analysis of the free molecule of base 12 has shown that the pyrazole and pyridine fragments are located in different planes (not coplanar)': This statement is not supported with data. Table 2 and Fig. 8 do not show the geometry of 12.
Response 12: In the Table 2 geometry of nucleosides, NOT base 12, is provided. Geometry of base 12 is shown now in Fig 8. The dihedral angles (C3-C4-C3Py-C4Py ) are included into Table 1.
Comments 13: Fig. 8: What do the red/green colors mean?
Response 13: Removed the color highlights.
Comments 14: Lines 326-327: 'two N1(2)-pyrazole tautomeric structures of compound 12': at pH 5-6? – it needs clarification, Which are N1 and N2? The structures should be numbered and referenced in the text.
Response 14: Structures are numbered in Table 1 and Fig. 8.
Comments 15: Paragraphs 326-357: the charges in the text are partially inconsistent with Fig. 8 (eg. -0.579, -0.5803, -0.5759) or Table 1 (-0.307). 'NH2 » -0.762 e' is not shown in Fig. 8.
Response 15: Recalculated and corrected.
Comments 16: Paragraph starting in line391: "In the NMR spectra of pyridinium nucleosides…' – the spectra are required to be shown here. At least 16, 17, and 19, with the discussed signals highlighted.
Response 16: Fragments of the 1H NMR spectrum of nucleoside 16 have been added to the Figure 9.
Comments 17: L. 399, 411, 415: '16' should be in bold
Response 17: Corrected.
Comments 18: Fig. 9: some "H" symbols are shown, some are omitted – and look like methyl groups.
Response 18: Corrected.
Comments 19: Fig. 9: why are some H's highlighted in green? The meaning of this should be explained in the caption.
Response 19: Removed the color highlights.
Comments 20: L. 482-483: something is wrong: 'Ошибка! Источник ссылки не найден'
Response 20: Corrected.
Round 2
Reviewer 3 Report
Comments and Suggestions for Authors
In the submitted revision, most of my comments were taken into account and corrected. However, I still have some concerns:
1. Extensive self-citation remained. It should be significantly limited.
2. Figure 6B should not be kept in this form. It is very unusual and is not in congruence with Fig. 7D/pH7. It may be just erroneous. It seems that after 24 h something happened to the enzyme(s) and 19 started to decompose. Such doubtful results should not be published. Strange results require repetitions.
The decrease in the amount of 18 between 1 and 48/96 h in Fig. 7D/pH7 is much less pronounced and may be a result of not perfect integration. From Fig. 7D-E one may conclude that pyridine 2’-deoxynucleoside is a very unstable glycolization kinetic product, which isomerize rapidly into 18 (thermodynamic product) or undergoes a second glycosylation to 19, or decomposes to 12 and dUrd. Due to instability, it cannot survive HPLC and cannot be detected. Cpd 19 is more stable and can be registered. It forms from the intermediate pyridine 2'-deoxynucleoside as long as there is free base 12. Then, it loses one dUrd and converts into 18. Something similar to Fig. SI-3 (p. SI-17) but at a constant pH (I have noticed Fig. SI-3 after writing this comment).
To gain a better insight into the reactions, two additional charts should be added to Fig. 7 showing the amounts of Urd and dUrd in the reaction mixtures analyzed.
3. Table 1 and Fig. 8 do not show any tautomeric equilibrium, although this is discussed in the text. It was shown in the previous version of Table 1. The 2D and 3D structures at pH 5-6 and the structures at pH 7 in Table 1 are not the same: in the 3D ones, the proton is localized on one of the nitrogen atoms, and N1 and N2 atoms are not equivalent. The same is shown for pH7 in Fig. 8 (where for pH 7 "N1=N2" is obviously inconsistent with the =N-NH- bonding). Which structures were used for the calculations of charges?
I have doubts if there is a need for Fig. 8. It is almost exactly the same as the structures in Table 1 and does not provide any additional information.
4. In the SI there is an additional discussion on the structures and calculations (p. SI-17 and SI-18). It should be mentioned in the main manuscript in Section 3.2 and/or 3.3.
These issues should be corrected before accepting the manuscript for publication.
Minor errors to be corrected:
· l. 84: "azapurin" ® "azapurine"
· l. 275: "мono" – wrong font for "m"
· l. 321: "May be" ® "Maybe"
· l. 369: sp2 ® sp2 ("2" in superscript)
· SI, p. SI-17 & SI-18:
o Sp2 ® sp2 , "2" in superscript
o "cite" ® "site"
o Something is wrong with the caption: "Ribosylation of heterocyclic base 12 at the pH 5-9 values of the reaction medium diverse pH values" – delete "diverse pH values" (probably).
o "Figure SI-3" ® "Figure SI-5"
o Refs [24-29] look to be irrelevant. Ref [44], and maybe other?
o Numbers 16a/16b are partially not in bold (p. SI-17 and Table SI-1)
Comments on the Quality of English LanguagePlease, check for typos.
Author Response
First and foremost, we would like to thank you for your close attention to our manuscript. We are impressed. And we are grateful for the attempt to explain some facts that remain unexplained so far.
Comment 1:
In the submitted revision, most of my comments were taken into account and corrected. However, I still have some concerns:
Extensive self-citation remained. It should be significantly limited.
Response 1:
We removed five self-citation links and added one review [14].
We would like to keep the remaining references to our articles, as they contain data on the glycosylation of a wide range of different substrates and note cases of non-specific glycosylation.
Comment 2a
Figure 6B should not be kept in this form. It is very unusual and is not in congruence with Fig. 7D/pH7. It may be just erroneous. It seems that after 24 h something happened to the enzyme(s) and 19 started to decompose. Such doubtful results should not be published. Strange results require repetitions.
Response 2a:
We repeated the experiment presented at Figure 6B and confirmed the results obtained earlier. Additional points (48 and 72 hours) were added to the chart.
Comment 2b:
The decrease in the amount of 18 between 1 and 48/96 h in Fig. 7D/pH7 is much less pronounced and may be a result of not perfect integration. From Fig. 7D-E one may conclude that pyridine 2’-deoxynucleoside is a very unstable glycolization kinetic product, which isomerize rapidly into 18 (thermodynamic product) or undergoes a second glycosylation to 19, or decomposes to 12 and dUrd. Due to instability, it cannot survive HPLC and cannot be detected. Cpd 19 is more stable and can be registered. It forms from the intermediate pyridine 2'-deoxynucleoside as long as there is free base 12. Then, it loses one dUrd and converts into 18. Something similar to Fig. SI-3 (p. SI-17) but at a constant pH (I have noticed Fig. SI-3 after writing this comment).
Response 2b:
Yes, we absolutely agree with you. However, firstly, it is possible to discuss the stability of the "invisible" product, but it is useless: until we have isolated it and studied its properties, nothing can be said. Secondly, it can be assumed that pyridinium deoxyriboside is not synthesized at all in the active site of the enzyme. Maybe, main pyrazole product 18 is subjected to the secondary glycosylation: in an hour the reaction mixture contains 80-90% of 18 and practically no base 12. But it contains a lot of 1-phosphate-2-deoxyribose (nearly 8-fold excess of moles).
We are now trying to change the conditions of the transglycosylation reaction so as to obtain pyridinium 2-deoxyriboside, isolate, confirm the structure, and then see at what rate it will be reglycosylated in the active site of the enzyme. But this is a separate investigation. Hopefully, we will be able to solve the mystery of the elusive nucleoside.
Comment 2c:
To gain a better insight into the reactions, two additional charts should be added to Fig. 7 showing the amounts of Urd and dUrd in the reaction mixtures analyzed.
Response 2c:
No, this cannot be determined by the amount of Urd, 2’-dUrd, and Ura, as they react in a 10-fold excess with the target base 12. We would like to note that the equilibrium of the reaction is provided by a system of two enzymes.
Comment 3a:
Table 1 and Fig. 8 do not show any tautomeric equilibrium, although this is discussed in the text. It was shown in the previous version of Table 1. The 2D and 3D structures at pH 5-6 and the structures at pH 7 in Table 1 are not the same: in the 3D ones, the proton is localized on one of the nitrogen atoms, and N1 and N2 atoms are not equivalent. The same is shown for pH7 in Fig. 8 (where for pH 7 "N1=N2" is obviously inconsistent with the =N-NH- bonding). Which structures were used for the calculations of charges?
Response 3a:
The term "tautomeric" is removed from the text, the charges are refined, and 2D structures are corrected. Everything was checked in the text. Unfortunately, we found errors in the calculation of the structures of base 12 associated with Ser90 and Asp204. Calculations take a long time and the data is not ready, so we deleted this data from Table 1 and changed the signature under Figure SI-19.
Comment 3b:
I have doubts if there is a need for Fig. 8. It is almost exactly the same as the structures in Table 1 and does not provide any additional information.
Response 3b:
Figure 8 is deleted.
Comment 4a:
In the SI there is an additional discussion on the structures and calculations (p. SI-17 and SI-18). It should be mentioned in the main manuscript in Section 3.2 and/or 3.3.
Response 4a:
Added to the main text "Figure SI-19 shows the equilibrium state of all the components involved in the base 12ribosylation process" (lines 409-410)
Comment 4b:
These issues should be corrected before accepting the manuscript for publication.
Minor errors to be corrected:
- l. 84: "azapurin" ® "azapurine"
- l. 275: "мono" – wrong font for "m"
- l. 321: "May be" ® "Maybe"
- l. 369: sp2 ® sp2 ("2" in superscript)
- SI, p. SI-17 & SI-18:
o Sp2 ® sp2 , "2" in superscript
o "cite" ® "site"
o Something is wrong with the caption: "Ribosylation of heterocyclic base 12 at the pH 5-9 values of the reaction medium diverse pH values" – delete "diverse pH values" (probably).
o "Figure SI-3" ® "Figure SI-5"
o Refs [24-29] look to be irrelevant. Ref [44], and maybe other?
o Numbers 16a/16b are partially not in bold (p. SI-17 and Table SI-1)
Response 4b:
Corrected